# Prevalence of child maltreatment in India and its association with gender, urbanisation and policy: a rapid review and meta-analysis protocol

Gwen Fernandes [1], Megan Fernandes,[2] Nilakshi Vaidya,[3] Philip De Souza,[4] Evgeniya Plotnikova,[5] Rosemary Geddes,[5] Bharath Holla,[3] Eesha Sharma,[6] Vivek Benegal,[3] Vikas Choudhry[7]

GF and MF are joint first authors.

For numbered affiliations see end of article.

**Correspondence to**
Dr Gwen Fernandes;
gwen.fernandes@bristol.ac.uk

## ABSTRACT

**Introduction** India is home to 20% of the world's children and yet, little is known on the magnitude and trends of child maltreatment nationwide. The aims of this review are to provide a prevalence of child maltreatment in India with considerations for any effects of gender; urbanisation (eg, urban vs rural) and legislation (Protection of Children from Sexual Offences (POCSO) Act 2012).

**Methods and analysis** A rapid review will be undertaken of all quantitative peer-reviewed studies on child maltreatment in India between 2005 and 2020. Four electronic databases will be systematically searched: PubMed, EMBASE, Cochrane and PsychInfo. The primary outcomes will include all aspects of child maltreatment: physical abuse, sexual abuse, emotional abuse, emotional neglect and physical neglect. Study participants will be between 0 and 18 years and will have reported maltreatment experiences using validated, reliable tools such as the Adverse Childhood Experiences Questionnaire as well as child self-reports and clinician reports. Study selection will follow the Preferred Reporting Items for Systematic Reviews and Meta-Analyses guidelines, and the methodological appraisal of the studies will be assessed by the Newcastle-Ottawa Quality assessment scale. A narrative synthesis will be conducted for all included studies. Also, if sufficient data are available, a meta-analysis will be conducted. Effect sizes will be determined from random-effects models stratified by gender, urbanisation and the pre-2012 and post-2012 POCSO Act cut-off. $I^2$ statistics will be used to assess heterogeneity and identify their potential sources and $\tau^2$ statistics will indicate any between-study variance.

**Ethics and dissemination** As this is a rapid review, minimal ethical risks are expected. The protocol and level 1 self-audit checklist were submitted and approved by the Usher Research Ethics Group panel in the Usher Institute (School of Medicine and Veterinary Sciences) at the University of Edinburgh (Reference B126255). Findings from this review will be disseminated widely through peer-reviewed publications and in various media, for example, conferences, congresses or symposia.

**PROSPERO registration number** CRD42019150403.

## Strengths and limitations of this study

► This is the first rapid review of the peer-reviewed and published evidence on child maltreatment in India drawing on studies from four electronic databases: PubMed, EMBASE, Cochrane and PsychInfo.

► The primary outcome is any type of prospectively re-called child maltreatment including physical abuse and neglect, emotional abuse and neglect and sexual abuse.

► First study to evaluate the effects of the legislative Act pre-2012 and post-2012 (Protection of Children from Sexual Offences Act 2012).

► The review is limited by the exclusion of the grey literature and non-English peer-reviewed publications due to the lack of resources and funding available to conduct the review.

► The review will include a narrative synthesis of non-governmental reports, media articles and other sources of grey literature to inform the findings.

## INTRODUCTION

In the past two decades, significant progress has been made in reducing child deaths. UNICEF estimates that, worldwide, there were 12.6 million children who died prematurely before the age of 5 in 1990 and this has declined to 5.3 million in 2018.[1] Of the children who do survive, 200 million do not reach their full potential due to a combination of family socioeconomic status, geographic location, ethnicity, disability or religious orientation.[1] It is every child's right to develop and thrive, not just survive. Balanced nutrition, good healthcare and consistent care and encouragement in supportive home and school environments and positive neighbourhood environments enable children to lead healthy lives.[2] The early years of childhood therefore provide the basis of intelligence and learning, personality and self-motivation,

social behaviour and collaboration, which extend into adulthood.[3]

The WHO defines child maltreatment as 'the abuse and neglect that occurs to children under 18 years of age and includes all types of physical and/or emotional ill-treatment, sexual abuse, and neglect which results in harm'. The WHO reports that worldwide, child maltreatment is widespread with one in four adults reporting being physically abused as children.[4 5] Research suggests that children who have experienced such adversity do not receive the right nutrition, care or opportunities to learn and grow as children in more affluent geographical areas or economies.[6 7] However, these findings are not generalisable to low/middle-income countries (LMICs) like India where millions of children may be exposed to other, specific adversities such as conflict, political instability, community violence and other traumatic life experiences.[7 8] Research on early childhood adversity and consequences has been limited in India, including a lack of cross-sectional and longitudinal studies.[9] These adversities such as abuse or neglect, along with associated habits such as underage smoking or drinking, formed in childhood, have been linked to a range of poor physical and mental health outcomes throughout the life course and ultimately greater premature mortality.[10] These adversities and consequent health inequalities can be exacerbated by socioeconomic status and poverty, urbanisation (rural areas or slum areas compared with urban zones) or even caste.[11]

In her book, *Bitter Chocolate*,[12] Virani discusses the context of these adversities in India—a highly patriarchal culture in which reprimanding, punishing or spanking a child (physically abusive behaviour) is a cultural norm.[13] The Indian economy is further characterised by external adversities from a rapidly growing economy, stark urban–rural divides, religious strife and geopolitical upheavals.[14] Data from UNICEF and the World Bank[15] show a global perspective of child rights and Indian children are marked as experiencing a difficult situation as categorised

by their rights, access to education and healthcare and exposure to child labour and child marriages (figure 1). It is no surprise that physiological and neurobiological studies have found that adversities such as child maltreatment can lead to changes in neural networks, impaired nervous, endocrine and immune systems development, resulting in physical, mental and emotional dysfunction and chronic physiological damage.[16–18]

### Indian context

In India, home to 19% of the world's children, it is estimated every second child is exposed to sexual abuse and violence.[19] The Indian National Crimes Records Bureau (NCRB) reports a child is sexually abused every 15 min and 53% of children report abuse by a parent, relative or school teacher.[20] The prevalence of child sexual abuse (CSA) in high-income countries is 20% for females and 8% for males[21] but in India, the estimates vary between 4%–66% for females and 4%–57% for males.[22–24] These figures are considered underestimates because of limited surveillance of direct childhood adversity such as physical abuse or emotional abuse as well as CSA due to stigma particularly affecting girls; under-reporting of cases by healthcare and police authorities; stigma and taboo associated with being a victim or experiencing abuse, and varying prevalence rates by geography (eg, urban vs rural vs slum settings) as well as varying communities (school-based, hospital-based, trafficking victims, etc).[24–27]

Poverty, and especially urban poverty in India is prevalent due to high economic migration that has translocated poverty from rural settings to urbanised areas.[28] While the proportion of deprived populations in India dropped from 45% to 22% between 1994 and 2012, one in every six urban Indians live in slums.[29] Health inequalities that exist as a result of poverty and rapid urbanisations need to be reduced so that child maltreatment, violence against girls and women, aims of the United Nations Sustainable Development Goals can be achieved. Bywaters *et al*[30] report strong associations between the

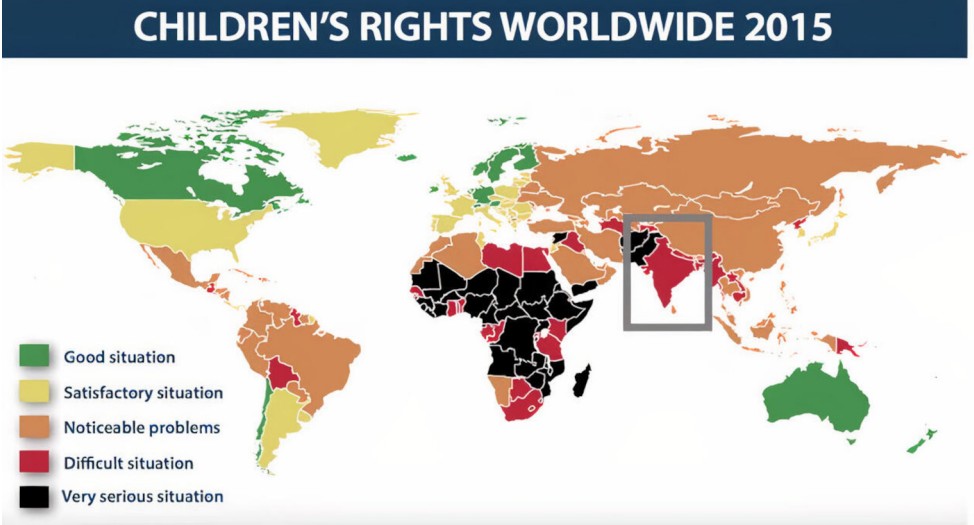

**Figure 1** World map showing the rights of a child as measured by UNICEF and World Bank data.[15]

socioeconomic status of a family and the risk of children within the household experiencing abuse and neglect. The greater economic hardship in areas like urban slums, the greater the likelihood and severity of child maltreatment. By contrast, we see evidence of the reverse in Deb and Modak's[31] study where violence is proportional to economic status as well as cultural beliefs and practices. Their data on socioeconomic status found that children from high-income households were almost four times more vulnerable to physical violence compared with children from low-income households. This is attributed to parents within high-income households having high expectations of their children especially with regards to academic achievement and a general undermining of a child's status compared with elder perspectives.[32] We see the nuanced effects of economics and urbanisation as children from more deprived backgrounds have parents who remain indifferent to their care and well-being especially for female children.[33] Given these findings, this review will carefully consider socioeconomic status of families, urbanisation metrics and specifically, the types of maltreatment affecting each sex within these settings.

Previous research has demonstrated that experiences of child maltreatment rarely occur in isolation and in fact, have a tendency to cluster.[9] Community-based studies have reported the prevalence and overlap of childhood sexual, physical and emotional abuse, but in India this has been limited. Choudhry et al's[22] systematic review focused on one aspect of child maltreatment: CSA, and was seminal work in detailing the high prevalence rates of CSA in Indian children. However, the authors acknowledged that CSA does not occur in silos and often co-occurs with other forms of child maltreatment in the same child.[34]

## Childhood adversity in India

Over the past 20 years, the phrase 'adverse childhood experiences' or (ACEs) have been studied as forms of maltreatment and household dysfunction affecting children either directly or indirectly. Felitti and colleagues[35] found a strong graded relationship between exposure to abuse or dysfunction during childhood and multiple risk factors for mortality in adulthood. ACEs have been associated with non-communicable conditions such as cardiovascular disease, cancer or mental illness.[7 36–38]

While alternative discourse on ACEs exists,[39] there are limitations in the application of this ACEs framework including different types of exposures, lack of consideration for frequency or severity of occurrence and the relevance to population level, rather than individual level, policies. Also, the application of this framework within LMICs like India may be limited given the effects of other adversities such as poverty, community violence and political strife.[40]

Using Felitti's ACEs framework and extending it with additional adversities,[40] figure 2 depicts different types of adversities each bearing significant ramifications extending from childhood to adulthood and beyond—affecting the family and society as a whole.[41] However, in India, the data on child maltreatment across various different states and settings are available but the data on ACEs are limited. A few recent studies have been able to identify the prevalence of ACEs in specific communities.[22] Damodaran and Paul[9] compiled data on ACEs and specifically, child maltreatment in youth based in Kerala (South India) and reported that in a sample of 600 young people, 91% had reported at least one ACE and over half the population reported three or more ACEs. However, in terms of severity, males reported exposure to more

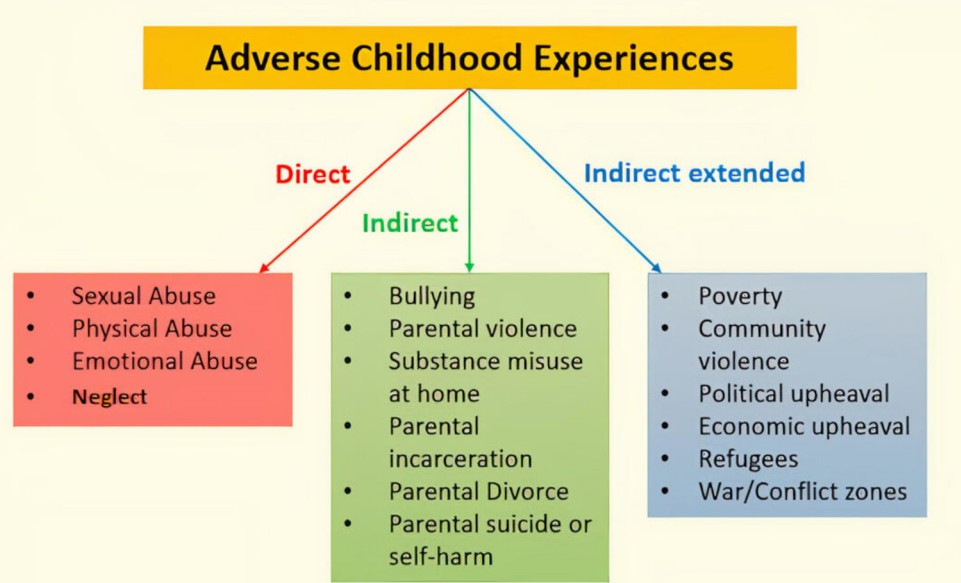

**Figure 2** Adverse childhood experiences (ACEs) including traditional direct ACEs (child maltreatment: physical, sexual and emotional abuse and neglect) and indirect ACEs (Felitti et al[35]) and further extended by researchers since the original study (Houtepen et al[40]).

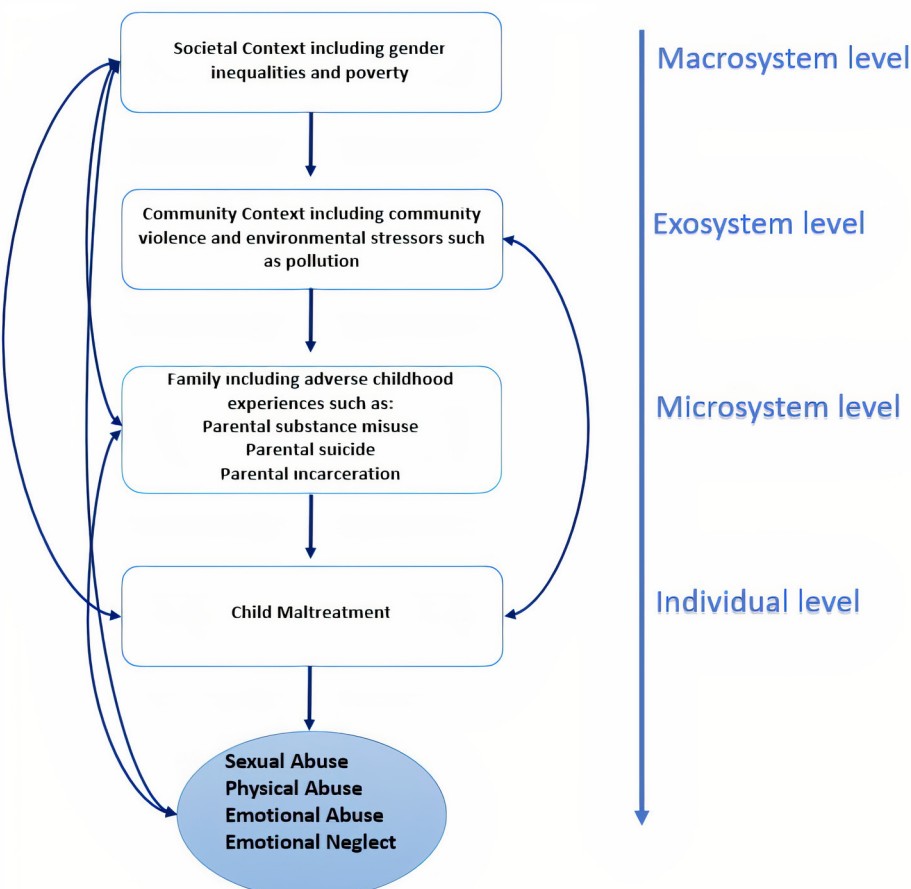

**Figure 3** Potential pathways explaining child maltreatment using the socioecological model proposed by Sidebotham and Heron.[45]

severe child maltreatment than females such as physical and sexual abuse.

In India, lower incomes, poor living conditions and living with domestic violence are adversities that can lead to neglect and abuse and often 'cluster'.[42] Figure 3 combines the socioecological model of child maltreatment[43] and suggests potential pathways (eg, from a macrosystem/societal or exosystem/community level) that may contribute to child maltreatment.[44 45] Particularly, around the individual level factors—there is a gap in the rates of other types of child maltreatment associated with CSA (eg, emotional abuse or neglect). While the topic of ACEs in India has received limited attention, CSA has received considerable attention from academics, clinicians, policymakers and legislators in recent years.[22 46] With the high prevalence of all types of child abuse in India, an attempt to amalgamate and compile the findings and present these figures in a systematic, cohesive and coherent manner is pressing.[5]

### Effect of gender
Previous research has suggested the nuanced effect of gender on the occurrence of child maltreatment in India.[22] In India, gender identification is illegal and yet, almost half of girls 'wish they were born boys'.[47] Although anecdotal evidence and case reports suggest that prevalence of sexual abuse is higher in female children than in male children, based on hospital services or care services utilisation,[48] some studies shows equal rates of sexual abuse across gender.[47 49] Deb[50] suggests that boys may be experiencing and reporting more sexual abuse while girls are less likely to be aware of abuse (knowledge and attitudes) or less likely to report abuse for fear of shame or stigma. Society is more dismissive of male abuse than that of girls as patriarchal tendencies to 'protect' include girls, but do not extend to boys.[49] However, when it comes to neglect with regards to food and education security, girls are known to be disadvantaged.[51] India is the only country where girls have greater rates of under-five mortality compared with boys. Indian girls are breastfed for shorter periods and consume less milk.[51] The reasons for this discrepancy are varied: there is a preference for sons over daughters in India and as an extension, more opportunities with regards to food, healthcare and education are afforded to male children.[52] However, given the complexities of gender effects in different types of abuse, generalisations cannot be drawn due to the limited number of studies, methodological limitations (eg, small sample sizes, non-generalisability of findings to the wider Indian

diaspora), urbanisation and socioeconomic privilege (eg, remote tribal areas being worse affected than wealthier urban ones[53]) and reporting bias (eg, particularly if parents reported via food diaries what was consumed).

## Effect of urbanisation

The prevalence of child maltreatment by urbanisation (rural, slum or urban areas) has received limited attention in the evidence base. Only two studies[54 55] included community populations from rural areas as well as urban areas. While other cross-sectional studies have been conducted in urban areas of India,[56] the child maltreatment prevalence figures (32% of boys and 42% of girls) are higher but not comparable due to differences in sample size; socioeconomic status; access to healthcare and support systems in urban areas compared with rural areas, as well as differing definitions of child maltreatment.

Banerjee[57] discussed that child maltreatment was more prevalent in metropolitan cities than in rural areas of India and their study found a high frequency of physical abuse among the slum and pavement dwellers although the severity of maltreatment was low. The study highlighted child labour practices (with 78.4% of children between the ages of 4–5 years working part-time jobs) as a result of living in a deprived area of the city. This study also highlighted the fact that often child maltreatment is a product of dire economic distress and within this context; small-scale regional or local studies might help understand the prevalence of child maltreatment in specific areas by urbanisation metrics. Over half the children who were classed as homeless (ie, living in drainage pipes, under tarpaulins, flyovers and those living in the open) reported CSA.[58] While child maltreatment figures such as CSA may seem high in urban or slum areas, the disadvantage with rural areas may be that abuse may go under the radar. They may be under-reported or disguised within local traditions such as a dowry culture or child marriages, all behaviours rooted in poverty and volatile socioeconomic circumstances.[59]

## Effect of Protection of Children from Sexual Offences Act

The Protection of Children from Sexual Offences Act (POCSO Act) was established in 2012 by the Indian government to protect children against offences like sexual abuse. The Act makes it mandatory to report acts of sexual offence and document reporting timelines, prosecution timelines and clearly record all evidence (within a 30-day period). While POSCO has been a relatively recent Act, its effect on number of reported offences has been debated.[22] While the gold standard remains as a multi-disciplinary, multi-faceted and holistic team providing psychological and legal support to the child and their families, this level of comprehensive care *under one roof* is yet to exist in India.[60 61] The impact of this study on policymaking will come from its inclusion of the only legislative act in India that focuses on CSA. Given the overlap between different types of abuse and neglect and their tendency to 'cluster', we will determine any effects of POCSO pre-2012 and post-2012 on all forms of child maltreatment. This has not been evaluated by the literature to date but was highlighted for future work by Choudhry and colleagues.[22]

## Rationale

There are known gaps in the evidence base with regards to the overlap of different forms of child maltreatment and specifically, the gendered effects of child maltreatment. These include the identification and motivation of perpetrators and what happens after maltreatment has happened—a child's awareness of their rights and attitudes towards abuse and neglect and reporting of child maltreatment. The circumstances that encourage reporting and details on protective factors and risk factors of child maltreatment have received limited attention in the literature.[22]

This review will build on these thematic and methodological gaps in the literature, synthesise the evidence by gender, urbanisations and POCSO effects and provide practice, policy and clinical solutions that may help reduce the occurrence and impact of child maltreatment in India.

## Research aims and objectives

The primary research aim is to describe the prevalence of child maltreatment in India. The secondary aim is to determine the effect of the POCSO Act on CSA and on other forms of child maltreatment.

The specific objectives were: to determine the prevalence of child maltreatment (physical abuse, sexual abuse, emotional abuse and physical and emotional neglect); to present rates by gender, urbanisation (rural, slum or urban) and study sample types (eg, community, school-going, high-risk such as trafficking victims, etc); to determine any impact of the POCSO Act (by the date of implementation of the Act, ie, 2012) on rates and to identify implications for future policy, practice and research amendments.

## METHODS AND ANALYSIS
### Study design

This review and meta-analysis will be carried out in compliance with the Cochrane Handbook for Systematic Reviews of Interventions (version 6.1) and reported in compliance with the Preferred Reporting Items for Systematic Reviews and Meta-Analyses (PRISMA) guidelines.[62–64]

### Patient and public involvement

It was not appropriate or possible to involve patients or the public in the design, conduct, reporting or dissemination plans of our research. The research team and advisory/expert panel consisted of clinicians, policymakers and research academics who work in the area of early childhood adversity and were able to advise on search terms used; databases selected and dissemination plans for the work on completion.

## Participants

### Inclusion criteria

Studies between 1 January 2005 and 1 January 2020 will be included to allow a comparison of an 8-year window pre-2012 and post-2012 POCSO Act. Studies with participants aged between 0 and 18 years will be included in the review; and will have experienced any form of child maltreatment as determined using International Society for the Prevention of Child Abuse and Neglect Child Abuse Screening Tool-Child Version, the Childhood Trauma Questionnaire, Adverse Childhood Experiences International Questionnaires (ACEIQ) or self-reports or clinician reports. ACEIQ or WHO definitions will also be used to determine child maltreatment[65] in addition to those by Day *et al*.[66] The primary outcome measure is any type of child maltreatment: sexual, physical or emotional abuse; and physical and emotional neglect. In addition to the definition, the time frame of recall, prospective recall in our case, warrants discussion.

Often, prospective and retrospective measures of child maltreatment do not always identify the same individuals.[67] Retrospective studies of child maltreatment are often regarded as a shortcut as significant time; finance and resources are not required to undertake prospective studies.[68] There is a low agreement between prospective and retrospective assessments of child maltreatment that can be explained by factors such as motivation of reporting, measurement features and memory bias. First, motivation can reduce agreement if individuals may gain something by intentionally withholding data about their maltreatment for embarrassment or fear or feeling uncomfortable or upset. Second, child maltreatment assessments are known to have imperfect test–retest reliability[69] and prospective measures might only account for the more severe of cases while retrospective 'true positive' case.[67] Finally, memory bias can reduce agreement as a result of impaired brain development,[70] deficit in memory consolidation of traumatic events or disrupted sleep patterns.[71] Therefore, given the variation, the focus on prospective recall (rather than retrospective) of child maltreatment for systematically reviewing the literature on abuse and neglect in children would reduce variation errors, measurement bias and memory bias.[67]

### Exclusion criteria

Studies that are qualitative in nature or mixed-methods (due to the time factor of this review), non-Indian study contexts, non-English studies, retrospective studies of adult CSA survivors, studies with insufficient peer-reviewed published data, media reports, newspaper articles and non-peer-reviewed articles will be excluded from this review. Only quantitative studies published in the English language will be included.

## Outcomes

### Defining childhood maltreatment

The WHO defines child maltreatment as 'the abuse and neglect that occurs to children under 18 years of age and includes all types of physical and/or emotional ill-treatment, sexual abuse, neglect which results in harm'. The National Society for the Prevention of Cruelty to Children (NSPCC) defines child maltreatment as harm experienced by a child by an adult or child *that is physical, sexual or emotional but can also involve a lack of love, care and attention*.[72] The NSPCC also set out clear descriptions of what constitutes child maltreatment and specifically each form of abuse and neglect that are summarised here.

### Child physical abuse

Physical abuse happens when a child is hurt or harmed deliberately causing injuries such as bruises, burns, cuts and broken bones. It may also include children being administered medication they do not need or not being given essential medicines for illness. The terms used specifically are kicking, shaking, slapping, throwing, poisoning or drowning. In India, these terms and additional ones such as 'spanking', 'rap' or 'pasting' also have equivalent meanings.

### Child sexual abuse

Sexual abuse happens when a child is forced or tricked into sexual activities. It happens when an abused makes physical contact with a child and includes touching, kissing and oral sex or non-contact abuse which includes exposing or flashing, exposing a child to sexual acts or making, viewing or distributing child abuse images or videos.

### Child emotional abuse

Emotional abuse includes acts that humiliate a child by constantly criticising them, insulting or shouting at them, blaming and making them a scapegoat or manipulating a child. These behaviours also include not allowing them to have friends or never saying anything kind or positive or congratulatory to a child. It perpetuates the feelings that a child is not loved, is worthless or inadequate.

### Child physical and emotional neglect

Neglect is the ongoing failure to meet a child's basic requirements and is often also classed as a form of child abuse. Physical neglect is when a child's basic needs such as food, clothing or shelter are not met or they are not properly supervised or kept safe. Emotional neglect is when a child is not provided with the nurture and stimulation they need to grow and thrive.

While child maltreatment is a global problem, it is certainly difficult to assess and manage in LMICs like India. Specific types of maltreatment such as abandonment, child labour, street-begging and corporal punishment are highly prevalent.[73] Therefore, while generic methods may exist to assess child maltreatment, the value of self-reports from children or from paediatricians, child psychologists or social workers and the police should not be dismissed.

**Table 1** Search term strategies using keywords and Medical Subject Heading (MeSH) terms

| Sexual abuse | Sexual abuse (cont) | Physical abuse | Emotional abuse | Emotional neglect | Additional child maltreatment definitions |
|---|---|---|---|---|---|
| Child* sexual abuse | Child abuse survivor | Child hitting | Child worthless | Denial clothing | Child neglect |
| Child sexual assault | Child sexual offender | Child shaking | Child unloved | Denial shelter | Child psychological assault |
| Child sexual coercion | Child pornography | Child throwing | Child inadequate | Denial food | Child denial |
| Child rape | Child molestation | Child poisoning | Child failure | Carer failure | Child pasting |
| Child prostitution | Child sexual crime | Child burning | Child unloved | Parental failure | Child emotional assault |
| Child crime victims | Child physical | Child scalding | Child corruption | Emotional harm | Non-accidental injury |
| Child incest | Child assault | Child drowning | Child unheard | Emotional danger | |
| Child perpetrator/ child victim 3 | Child beating | Child suffocating | Cruelty | Child* neglect | |
| Child physical violence | Child emo* abuse | | | Abandonment | |
| Child mental violence | Child maltreat* | | | Bereavement | |
| Child sexual violence | Child sexual coercion | | | Grief | |
| Child sexual offence | Child sexual aggression | | | | |

Search terms: ("child abuse, sexual"[MeSH Terms] OR ("child"[All Fields] AND "abuse"[All Fields] AND "sexual"[All Fields]) OR "sexual child abuse"[All Fields] OR "physical child abuse" [All Fields] OR "emotional child abuse"[All Fields] OR ("child"[All Fields] AND "sexual"[All Fields] AND "abuse"[All Fields]) OR "child sexual abuse"[All Fields]) AND ("india"[MeSH Terms] OR "india"[All Fields])—expanded from Choudhry et al[22] and Day et al[66] and updated to include child maltreatment definitions.
*Alternative search terms to include adolescent sexual abuse, adolescent physical assault and so on.

### Search strategy

The relevant search terms to be included in the search strategy will be identified during a preliminary search reported in the study protocol using PubMed's Medical Subject Heading's (MeSH) browser. The search contained both free-text terms and controlled vocabulary like MeSH terms as well as truncation and Boolean operators. In order to improve the quality of the search strategy and avoid bias or errors, the strategy includes choosing all appropriate terms, both descriptors and synonyms including those that may be specific to the Indian diaspora.[74] First, the initial search terms were developed with reference to seminal works or major publications of child maltreatment and early childhood adversity in India.[9 26 47 75] The review then used the MeSH terms and keywords documented (table 1) and were further enhanced by the search strategy from a systematic review conducted on CSA in Indian children.[22] These were further validated with input from an expert advisory panel in the field of systematic reviews and ACEs in India (online supplemental appendix 1). Five electronic databases will be systematically searched: PubMed, EMBASE, Cochrane and PsychInfo and Campbell Library. PubMed and EMBASE are the two largest healthcare bibliographic databases containing the vast majority of published health-related literature.[74 76] Searching Cochrane and PsychInfo will provide nuanced searches of other systematic reviews of child maltreatment or ACEs from the Asian continent and specifically from India. Systematic reviews will be included in the search to identify potentially relevant studies that may have been excluded by the electronic search strategy. Online supplemental appendix 2a–d contains the search strategies used in the four databases, respectively. These search strategies and advice on specific electronic databases will be cross-checked and verified by an information specialist. The final search strategy used will be published in detail in the future published manuscript of results.

### Data management

The selection of studies will be carried out systematically by two people independently, and in accordance with PRISMA guidelines, to improve accuracy and to minimise bias.[63] The first reviewer (first author) is an epidemiologist with expertise in early childhood adversities and the impact on health outcomes; the secondary reviewer is a clinical psychologist with expertise on ACEs in India while the third and fourth reviewers, are clinical doctors with expertise in general medicine and paediatric care, validated the selection of studies pre-2012 and post-2012 threshold.

All studies identified from the individual search engines will be imported into a reference manager, EndNote, which will be used for de-duplication. The references will then be extracted into Rayyan for management between the first author and secondary reviewers.[77] All abstracts and full-text articles will be reviewed independently by the first author and secondary reviewers as a fixed percentage of the work (40%). The details of all articles processed (including duplicates and details of excluded texts at each screening stage) as well as reasons for inclusion and exclusion from the study will be recorded and logged for the first author and secondary reviewers and presented in a PRISMA flowchart. At the title/abstract screening stage, the authors will use Rayyan's coding system (traffic light system) with appropriate tags such as 'wrong outcome of interest' in order to code each study (done independently by the authors on Rayyan's blinded review setting). Once all titles/abstracts are screened, we will unblind all authors and discuss those titles/abstracts to be included, or excluded or further discussed with the expert panel as a result of discrepancies.[77] All selected full-text articles will be screened by the primary author and secondary reviewers and the reasons for exclusion of these articles will be recorded and presented in the final manuscript.

### Data extraction
The following data will be extracted from the included studies: title; author information; year of study (not publication date); study design; recruitment strategy and sample characteristics (school-based, community, hospital based, etc); sample size; urbanisation metrics (urban or rural or slum); response rate; definition of child maltreatment used; timeframe of child maltreatment (in years, eg, past 1 year or past 6 months or half a year); mean age of participants; gender distribution; statistical methodology; confounder data including parental relationships, home ownership, family settings (eg, nuclear, joint, mixed), religion, educational status and literacy rates; prevalence rate of each type of child maltreatment; author's key message or conclusions; ethical approval and any conflicts of interest. An example of a completed data extraction form is presented in table 2.[26]

### Quality assessment
The final selected studies will be appraised for quality by the first author and, to ensure robustness, they will also be quality appraised by a second reviewer. A third reviewer will resolve any conflicts, with support from the expert advisory panel if required.

Study quality (largely cross-sectional and case–control) will be assessed using the Newcastle-Ottawa Quality assessment scale (NOS) adapted for use in cross-sectional study designs as it considers key areas: selection (representativeness of the sample, sample size, non-responders and exposure details); comparability (is confounding considered) and outcome (blinding, recording and statistical test used).[78 79] While there are a myriad range of assessment tools available, the NOS has been endorsed by the Cochrane Collaboration to assess quality of research studies.[76] An example of an NOS assessment of a sample study is presented in online supplemental appendix 3.

### Data synthesis
The study will include a narrative synthesis of the findings and attempt a meta-analysis dependant on heterogeneity. A meta-analytic approach will be used including any random-effect pooled prevalence of child maltreatment stratified by gender, urbanisation and pre-POCSO and post-POCSO Act. Heterogeneity will be evaluated using forest plots and measuring the impact of the $I^2$ statistic ($I^2 > 50\%$ indicates significant heterogeneity) and tested with the $\chi^2$ statistic ($p < 0.05$).[62] We will also report $\tau^2$ values to indicate any between-study variance. We will use a fixed effects model for meta-analysis, except where we identify statistical heterogeneity in which case, we would use a random effects Poisson model[80] to determine an 'average prevalence' across the population of all included studies. We will use the 'metaprop' command to perform meta-analyses of proportions of child maltreatment in STATA version 16, which builds on the 'metan' command[81] but is typically used to pool effects such as proportions from cross-sectional studies instead of ORs from case–control studies.

In the case of significant heterogeneity derived from our analysis, we will attempt to create subgroups of the paediatric population to understand any extracted data. This could include different age-bands of participants (eg, child (0–12 years), adolescent (13–18 years) using WHO cut-offs for age bands in paediatric populations. We may also wish to compare prevalence rates of child maltreatment based on self-report measures versus objectively assessed measures as well as compare rates by time (1 year compared with lifetime occurrence or frequency data).

### Meta-bia(s)
Research papers were written originally in Hindi or any other Indian dialect, that would otherwise be included in the data capture, will not be used for the purpose of this study.

### DISCUSSION
The findings of this study will aim to add to the limited knowledge of child maltreatment in India and specifically, the impact of gender, urbanisation and the POCSO Act on the prevalence and determinants of child maltreatment. There are known gaps in the evidence base with regards to the overlap of different forms of child maltreatment and specifically, the gendered effects of child maltreatment. These include the identification and motivation of perpetrators and what happens after maltreatment has happened—a child's awareness of their rights and attitudes towards abuse, neglect and reporting of child maltreatment. The circumstances that encourage

**Table 2** Example of a data extraction form

| | |
|---|---|
| Title of study<br>Authors<br>Year of study | 'Prevalence of child abuse in Kerala, India: An ICAST-CH based survey'<br>Manoj Therayil Kumar, Nilamadhab Kar, Sebind Kumar<br>2019 |
| Study design | Cross-sectional survey in Thrissur, Kerala |
| Recruitment strategy | There are 39 schools in Thrissur with 15 150 students in attendance. Random number generation in excel was used to select half the schools on the list and all students in grades 8–10 who attended school on that day were eligible to take part |
| Sample setting (school/college, clinical, community)<br>Sample size and response rate | School setting<br>6957 participants took part in the survey. No details of response rate were provided however the authors state that no questionnaire was returned completely blank |
| Child maltreatment definition used (timeframe defined) | All abuse-related questions in the ICAST-CH were scored on a severity scale and categorised into both 1-year prevalence and lifetime prevalence |
| Classification tool<br>validity/reliability of tool | The International Society for the Prevention of Child Abuse and Neglect (ISPCAN) Child Abuse Screening Tool-Child Home Version (ICAST-CH) was used<br>Developed in unison with the WHO and UNICEF and has good criterion validity and was translated into Malayalam and was piloted for use before being administered to the students. Internal consistency measures using Cronbach's alpha for subscales—exposure to violence, physical abuse, emotional abuse, neglect and sexual abuse |
| Urban/rural/slum-dwelling participants breakdown | Not mentioned |
| Mean age (SD) | 13.9 years (1.12) for males and 13.7 years (0.99) for females |
| Gender distribution | 2071 males (30.1%)<br>4810 females (69.9%) |
| Example: CSA prevalence by gender | 30.9% in males for 1-year prevalence<br>10.8% in females for 1-year prevalence<br>34.2% in males for lifetime prevalence<br>14% in females for lifetime prevalence |
| Statistical methods used | Frequency and counts used to determine prevalence and presented with 95% CIs<br>Cronbach alpha used to determine internal consistency of the Malayalam version of the scale<br>Significance was set at the 0.05 level as a maximum but 0.01 and <0.001 were reported |
| Confounders measured and/or adjusted for | Measured confounders—gender, age, religion, family type, accommodation, parental accommodation and socioeconomic status |
| Other confounders discussed but not measured | Literacy rates, cultural strata of participants |
| Participants experience of:<br>Emotional abuse (rate, %)<br>Physical abuse (rate, %)<br>Sexual abuse (rate, %)<br>Emotional neglect (rate, %)<br>Physical neglect (rate, %) | |
| Conclusion/comments | The abuse reported was more occasional experiences rather than frequent reporting nonetheless as 89.9% of any abuse in the past 1 year and 91% in the lifetime. First, study in Indian diaspora to use the ICAST-CH tool to estimate household abuse. Did not include external environments such as schools |
| Conflicts of interest | None declared |
| Ethical approval | Ethical approval was sought from local authority and educational ministry to conduct the research |
| Funders | Municipal Corporation of Thrissur city |

CSA, child sexual abuse.

reporting, details on protective factors and risk factors of child maltreatment have received limited attention in the literature.[22]

Furthermore, the impact of child maltreatment and where they occur (eg, urban areas vs slum areas) is important as these may offer an insight into potential interventional pathways that differ by geography, for example, enforcing child welfare policies in rural areas.[22] The impacts of urban and rural poverty, socioeconomic status; family income; parental occupation; family structure (nuclear, joint, extended, etc) and other confounder data will be collated during this review to aid in the interpretation of results including subgroup analysis and meta-analysis. Furthermore, the impact of this study on policymaking will come from its inclusion of the only legislative act in India that focusses on child maltreatment. The POCSO Act was established in 2012 by the Indian government to protect children against offences like sexual abuse. The Act makes it mandatory to report acts of sexual offence and document reporting timelines, prosecution timelines and clearly record all evidence (within a 30-day period). While POSCO has been a relatively recent Act, its effect on the number of reported offences in research studies has yet to be determined.

## Implications on policy and research

When the evidence on child maltreatment in India is collated and presented, any policy or practice impacts can be tailor-made based on the country-specific context and findings. In India, the challenge will come from addressing cultural norms and traditional values (eg, gender inequity and reducing child marriages), creating safer environments for children within homes and schools (eg, authorities penalising assault) and scaling up response and support services for children in different settings such as tribal regions, slum areas, rural communities (eg, trauma informed health and social care). This approach has worked in higher-income countries but in LMIC's like India, there are challenges. These include low levels of trauma-informed care for injured children in hospital settings with 91% of healthcare workers not receiving any trauma-informed training.[82] Other challenges included overburdened judicial and criminal systems; poor child welfare services; gender inequality and illiteracy.[22] This study will capture data on health inequities and social determinants that predispose to maltreatment in different geographical areas of India including confounders such as poverty and neglect.

## Strengths and limitations

There are some strengths to our approaches. This is the first review undertaken focussing on all aspects of child maltreatment in India where previous review articles and studies have focused on a specific facet of maltreatment, for example, CSA. Second, the inclusion of the POCSO Act makes this review nuanced to comment on any effect of legislation where it has never been formally evaluated.[83] This study will include an analysis of child maltreatment studies and corresponding prevalence rates, both pre-2012 and post-2012. With the inclusion of the POCSO 2012 threshold, it is expected that there will be an increased number of cases reporting all forms of child maltreatment. Third, the authors will attempt to meta-analyse the data depending on heterogeneity scores generated. Choudhry et al's work[22] highlighted the significant heterogeneity of studies conducted to date and this might limit the quality of meta-analytics possible given the discrepancies in samples size. Possible reasons for heterogeneity include varying age ranges of participants, selection bias with sample provenance (eg, recruitment from observation homes vs school children), varying urbanisation metrics (eg, samples including children from all types of socioeconomic backgrounds) and variation in outcome assessment definition (eg, self-report vs validated questionnaire).

There are limitations to this review. First, only papers in English were included due to limited access to funding and resources such as a professional translator. We acknowledge the importance of Anglophone bias in the context of this Indian based study therefore endeavour to ensure future extensions of this research does include non-English papers Second, excluding grey literature from OPEN Grey, narrative reports from non-governmental organisations or World Bank reports would limit some of the 'unmeasured' exposures and confounders that may increase child maltreatment.[25 49]

However, the authors will make attempts to collate this additional information so that they may be presented as part of a narrative synthesis on the topic of child maltreatment in India, thus improving the quality, generalisability and relevance of our work.[84] The exclusion of national statistics (eg, NCRB records) might limit interpretation of any effect of the POCSO Act on conviction rates for child maltreatment. Qualitative studies were also excluded from this review which may not effect prevalence rates of child maltreatment but would allow exploration of non-responses from children (interpretative) or indeed, highlight determinants that have not been identified in quantitative or aggregated reviews.[85–87]

This review will build on these thematic and methodological gaps in the literature, synthesise the evidence by gender, urbanisations and POCSO effects and provide practice, policy and clinical solutions that may help reduce the occurrence and impact of child maltreatment in India.

## Ethics

A level 1 self-audit checklist was approved by an ethics panels at the Usher Institute at the University of Edinburgh. Minimal ethical risks are expected. However, Vergnes et al,[64] propose that for systematic reviews, particularly on sensitive topics authors should consider a minimal ethical assessment review and report these in their research studies. As a result, in this review, ethical approval details from each included study will be collected.

## Dissemination

The findings of this review will be published in a peer-reviewed journal, and disseminated at national and international conferences and through social media.

**Author affiliations**
[1]Population Health Sciences, University of Bristol, Bristol, UK
[2]Department of Surgery, Norfolk and Norwich University Hospital NHS Trust, Norwich, UK
[3]Department of Child and Adolescent Psychiatry, National Institute of Mental Health and Neuro Sciences, Bangalore, Karnataka, India
[4]Department of Medicine, Norfolk and Norwich University Hospital NHS Trust, Norwich, UK
[5]College of Medicine and Veterinary Medicine, The University of Edinburgh Usher Institute of Population Health Sciences and Informatics, Edinburgh, UK
[6]Child and Adolescent Psychiatry, National Institute of Mental Health and Neuro Sciences, Bangalore, Karnataka, India
[7]Public Health, Sambodhi Research and Communications Pvt Ltd, Noida, Uttar Pradesh, India

**Correction notice**  Joint first authorship has been added for this article.

**Acknowledgements**  The authors would like to extend their sincere gratitude to Mrs Sarah Herring, Subject Librarian for Population Health Sciences, University of Bristol for her advice and guidance during the protocol development stage.

**Contributors**  GF, RG and EP were responsible for the initial protocol drafting and design. GF, MF, NV, PDS then contributed to preliminary searches used to develop the rationale and background of the study. ES, BH, VC, VB, PDS and MF were on the expert advisory panel for the study to advise on methods design and statistical analysis techniques that could be used. All authors contributed and approved this final manuscript.

**Funding**  GF is funded by the UK Medical Research Council and Alcohol Research UK Grant (MR/L022206/1).

**Map disclaimer**  The inclusion of any map (including the depiction of any boundaries therein), or of any geographic or locational reference, does not imply

the expression of any opinion whatsoever on the part of BMJ concerning the legal status of any country, territory, jurisdiction or area or of its authorities. Any such expression remains solely that of the relevant source and is not endorsed by BMJ. Maps are provided without any warranty of any kind, either express or implied.

**Competing interests**  None declared.

**Patient consent for publication**  Not required.

**Provenance and peer review**  Not commissioned; externally peer reviewed.

**ORCID iD**
Gwen Fernandes http://orcid.org/0000-0003-0203-7053

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
