## [Reviewer comments · BMJ Open]

ARTICLE DETAILS

TITLE (PROVISIONAL)	The prevalence of child maltreatment in India and its association with gender, urbanisation and policy: a rapid review and meta-analysis protocol
AUTHORS	Fernandes, Gwen; Fernandes, Megan; Vaidya, Nilakshi; De Souza, Philip; Plotnikova, Evgeniya; Geddes, Rosemary; Holla, Bharath; Sharma, Eesha; Benegal, Vivek; Choudhry, Vikas

VERSION 1 – REVIEW

REVIEWER	Xu, Yicheng Gansu university of Political Science and Law, School of Public Administration
REVIEW RETURNED	25-Dec-2020

GENERAL COMMENTS	The research goals are clear, the research method is appropriate, and the research conclusions are also convincing. It can fully reflect the current situation of India children's life at the level of India's economic development in the new era. The figures, tables, keywords, abstract, and references in the article all comply with standards. The results of the research are also expected to improve the overall condition of children in India, and the authors are expected to propose a better and generalizable evaluation framework.
--

REVIEWER	Hanratty , Jennifer Queen's University Belfast
REVIEW RETURNED	18-Jan-2021

GENERAL COMMENTS	Overall, the authors make a compelling case for the need for a review now. Unfortunately, the proposed review does not currently meet the standards that I consider necessary to be a systematic review. As I see it there are two options here. First, make the suggested improvements to the search and inclusion criteria to meet the standards for systematic review. The second option, if the author team do not have the resources to improve and extend the search, is to resubmit as a rapid or restricted review. Detailed feedback: Abstract: Line 16 should psychindex be Psyclnfo? Background: The authors make a compelling case for the need for their proposed review. I feel that the opening paragraph is a little muddled. For example in line 21-23 the authors refer to children who experience maltreatment compared to children in more affluent geographical areas, but child maltreatment is not exclusive to LMICs. The intention to examine the differential rates of child maltreatment (CM) by gender and urbanization is important. Have the authors
--

	considered using the PROGRESS+ checklist to identify other characteristics that may affect rates of CM? PROGRESS-Plus Cochrane Equity Methods: Study design: PRISMA provides guidance for the reporting of reviews but not their conduct, please change the wording to reflect this. I suggest consulting the Cochrane handbook for guidance on the conduct of the review or JBI handbook chapter 5, on conducting prevalence reviews. Chapter 5: Systematic reviews of prevalence and incidence - JBI Manual for Evidence Synthesis - JBI GLOBAL WIKI Participants: Will the authors only include studies where children are reporting on current abuse? Or will you include studies where children report on incidences of abuse in the past? If the latter than I suggest extending the upper age limit to include young adults (up to 25) so that studies that collect retrospective reports of relatively recent childhood experiences might be included. I would also like to know why retrospective studies of CSA will be excluded – is there a particular reason for this? What will the authors do with studies that include both children/adolescents and adults? E.g. 15-25 year olds? Outcomes: Some definitions of CM include witnessing domestic violence. Can you consider including this? Search: Overall, I do not think that the search strategy is adequate and I suggest that the team consult an information retrieval specialist to improve the search strategy. Obvious terms are missing such as “non accidental injury”, “child maltreatment”. An example search that the team might consider using, developed and tested by an info specialists, is available here (lines 1-20) https://www.ncbi.nlm.nih.gov/books/NBK385397/ The Cochrane library predominantly consists of intervention reviews and RCTs and is unlikely to contain reviews or primary studies relevant to this review. I suggest searching databases specifically for relevant Systematic Reviews (the search strings developed by Sandieson et al are excellent https://sites.google.com/view/pearl-harvesting-search/ph-synonym-rings/structure-or-study-design/systematic-reviews In addition to searching the databases already listed in the protocol more fruitful systematic review databases for this topic would be the Campbell Library/Journal and JIB https://journals.lww.com/jbisrir/pages/default.aspx PsychIndex – is this an error? Presumably, the authors mean PsycInfo? The team have not indicated that they intend to search for grey literature which is major omission. I suggest adding, at a minimum, web searches to identify grey literature sources such as government reports, NGO data, international surveys that include India etc. I see later in the discussion that this omission is due to limited resources. I suggest that this makes this work a rapid or restricted review rather than a systematic review.
--	--

	Data Management: The data management process is not sufficiently clear in my opinion. Will dual independent screening be done at title/abstract screening stage and full text? Recording reasons for exclusion and inclusion for all studies is not typically necessary; instead I suggest reasons for exclusion only be recorded for articles screened at full text stage. Quality assessment: This section conflates screening quality with assessment of quality and risk of bias in included studies. Please separate the two. The quality assessment section should only refer to how risk of bias/quality of the studies included in the review will be assessed. The Newcastle Ottawa Scale (NOS) was primarily designed to assess quality of non-randomised studies of outcomes and NOT cohort/cross-sectional studies of prevalence. Adapted versions of the scale are available and should be considered. Please document any adaptation to the scale used. Can the authors state why they will exclude studies written in languages other than English? Will the impact of this decision be discussed in the review? Discussion: Typically, a protocol would not include a discussion and some of the information here belongs in the methods section, specifically Patient and public involvement: I do not agree that it is not appropriate for the public to be involved in the conceptualization of systematic reviews. PRISMA: While the authors have filled out the PRISMA checklist the protocol does not actually contain enough detail to meet PRISMA standards. E.g. Was a protocol already produced with PROSPERO? The authors have not specified eligibility criteria in enough detail. Specifically are there limits on the year the study was conducted? What will you do with studies including participants with an overlapping age range? Information sources are not described with dates, errors are present and potentially useful databases are not included (e.g. CINAHL, Web of Knowledge SSCI, Campbell Library, JBI). The search is not adequately developed or reported. I commend the team on a good idea for a review that is needed and hope that my comments are helpful in improving the quality of the work.
--	---

REVIEWER	Walsh, David Glasgow Centre for Population Health
REVIEW RETURNED	18-Jan-2021

GENERAL COMMENTS	Thanks for the opportunity to comment on the protocol for a systematic review of child maltreatment in India. Overall I found the protocol to be helpfully detailed in terms of its rationale and proposed methods. My comments, therefore, are all relatively minor, with the sole aim of hopefully adding some additional clarity to a small number of aspects. The comments are as follows:  1. Other than a passing mention of confounders in the data collection Table (Table 2), there is no real discussion of poverty/socioeconomic position, which I found rather surprising. Given the clear understanding of the role of poverty in increasing
---

risk of maltreatment (e.g. see the JRF review from 2015 among other papers), I would have thought this would at the very least have been flagged up as a key issue in the review i.e. the need to collect such data to aid interpretation of results (including of any meta-analysis).

2. It would also be helpful to justify why the authors are only looking at papers published between 2005-2020. I suspect (?) it relates to the date of the child sexual abuse legislation that is mentioned, but it's not clear. As a 'by the by', the inclusion criterion is stated in the abstract, but not (I don't think?) in the actual methods, and obviously should be (although apologies if I have just missed it!)

3. Additionally regarding dates, reviewers of protocols are specifically asked to check that the dates of when the study is being/will be undertaken are included, so these could also be helpfully added.

4. The methods also state that retrospective studies will be excluded. Given that quite a few studies in the literature are studies of adults asked about maltreatment in childhood, it would be good just to justify/explain this exclusion.

5. Non-English language papers are also excluded. Although understandable, it would be helpful – particularly for a study of maltreatment in India – to say a little bit more about that (again in terms of justification). As the authors will know, the issue of 'Anglophone bias' is often highlighted, and your study is one where I would have thought careful and clear justification for excluding non-English papers would be particularly helpful.

6. In terms of the meta-analysis, the authors may find it helpful to think about (and discuss) how any such analysis might deal with profound differences in measurement across different studies e.g. of age of child (clearly important in terms of potential exposure to maltreatment), or of the number and types of maltreatment recorded (e.g. comparing studies of sexual abuse alone, with studies of multiple forms of maltreatment).

7. There is no mention of whether or not any grey literature will be included in the review, and it would be useful to be clear about this also.

8. Finally, this is a review of child maltreatment, as is clear from both the title and stated study design outcomes. However, a large chunk of the introduction discusses the much broader concept of adverse childhood experiences (ACEs) – and I'm not sure how helpful that is. As all the authors will know well, maltreatment accounts for around half the often-used collective measure of 10 ACEs, with the other half of the latter being a much broader set of measures of (unhelpfully titled) "household dysfunction". Although ACEs have obviously been the focus of a huge amount of research in recent years (see critique by Kelly-Irving and Delpierre regarding reasons why), they have also subject to considerable criticism for a number of reasons. There is also a large, separate, literature on maltreatment which – again as you will know – predates much of the ACEs research. I think it would be clearer, therefore, just to focus on maltreatment alone, without the potentially confusing mentions of ACEs (other than stating that

	maltreatment is included within some larger questionnaires including those examining ACEs). For the same reason, I'm not sure the term "maltreatment ACEs" is particularly helpful in that regard. Finally (no. 2), but most importantly, a systematic review is often a hard shift – so I wish you luck with what is clearly a very worthwhile and important endeavour.
--	--

VERSION 1 – AUTHOR RESPONSE

Reviewer: 1

Dr. Yicheng Xu, Gansu university of Political Science and Law

Comments to the Author:

The research goals are clear, the research method is appropriate, and the research conclusions are also convincing. It can fully reflect the current situation of India children's life at the level of India's economic development in the new era. The figures, tables, keywords, abstract, and references in the article all comply with standards. The results of the research are also expected to improve the overall condition of children in India, and the authors are expected to propose a better and generalizable evaluation framework.

Response to Reviewer 1, Dr Yicheng Xu:

Thank you, Dr Xu, for your constructive feedback. We believe that this review will attempt to reflect the current situation of child maltreatment prevalence in India. It is a time when India is undergoing significant economic and social change and as a result, these findings will no doubt give us the scale of the issues over the past 15 years and also highlight ways and means we may improve address maltreatment and the consequences at a population level.

Reviewer: 2

Dr. Jennifer Hanratty , Queen's University Belfast

Comments to the Author:

Overall, the authors make a compelling case for the need for a review now. Unfortunately, the proposed review does not currently meet the standards that I consider necessary to be a systematic review. As I see it there are two options here. First, make the suggested improvements to the search and inclusion criteria to meet the standards for systematic review. The second option, if the author team do not have the resources to improve and extend the search, is to resubmit as a rapid or restricted review.

Response to Reviewer 2, Dr Jennifer Hanratty:

Thank you, Dr Hanratty, for your detailed feedback on our protocol paper. We have revised our manuscript as per your suggestions above by using a combination of approaches – improving our methodology, clearly detailing the limits or constraints of this work, and resubmitting as a rapid review. We have detailed this below and within our manuscript as marked changes.

Detailed feedback:

1. Abstract: Line 16 should psychindex be PsycInfo?

Line 16 (now 44) of our abstract refers to the electronic databases that will be searched as part of this review. The authors confirm that the database will be PsychInfo as this is the database, we have access to at the University of Bristol. We have also confirmed this with our information specialist and subject librarian (covering Population Health Sciences) who will be providing us with advice with this project (Mrs Sarah Herring, University of Bristol). Therefore, we have made the corrections through the manuscript.

2. Background: The authors make a compelling case for the need for their proposed review. I feel that the opening paragraph is a little muddled. For example in line 21-23 the authors refer to children who experience maltreatment compared to children in more affluent geographical areas, but child maltreatment is not exclusive to LMICs.

We thank the Dr Hanratty for her comments on child maltreatment in different settings. Indeed, the literature has repeatedly shown that child maltreatment is not exclusive to LMICs and it is indeed a global issue (WHO 2020). However, it has been acknowledged by several experts in the field that the majority of the literature on adverse childhood experiences and their impact are focussed in higher income countries which tend to be better resourced, more nuanced and more reflective of those specific populations (Bellis et al., 2019; Blum et al., 2019; Manyema and Richter 2019). The adverse childhood experiences however are not generalisable to lower-and middle-income nations where millions of children are exposed to conflict, community violence and other traumatic life experiences (Bellis et al., 2019; Sharma et al., 2020). We have drawn on existing data from within specific regions or cultural contexts to address the scope and impact of child maltreatment within a specific country or region. In addition, the impact of poverty or specifically nuanced gender bias, may further exacerbate abuse and neglect and the impact on children's health and well-being.

As a result, we would like to retain our original introduction in framing child maltreatment as a firstly, a global issue, but also, as a specific issue to LMIC nations such as India where individual investigation into prevalence is warranted and pressing. We have expanded our introduction to accommodate suggestions made by Dr Hanratty and subsequently, by Dr Walsh (Page 3, Lines 105-108).

3. The intention to examine the differential rates of child maltreatment (CM) by gender and urbanization is important. Have the authors considered using the PROGRESS+ checklist to identify other characteristics that may affect rates of CM? PROGRESS-Plus | Cochrane Equity

We thank Dr Hanratty for her suggestion on capturing additional information from the evidence based in addition to gender, socioeconomic status and urbanisation rates. The authors confirm we will also capture additional information (where available) from individual research studies such as educational status and literacy rates or social capital as well as positive factors such as parental relationships, home ownership and family settings. We did include this in our original manuscript as part of a table (Table 2 on data extraction, pgs12-13) but have now included a segment on this with references in the Methods (Page 13, Lines 537-551).

4. Methods:

Study design: PRISMA provides guidance for the reporting of reviews but not their conduct, please change the wording to reflect this. I suggest consulting the Cochrane handbook for guidance on the conduct of the review or JBI handbook chapter 5, on conducting prevalence reviews. Chapter 5: Systematic reviews of prevalence and incidence - JBI Manual for Evidence Synthesis - JBI GLOBAL WIKI

We have taken on board the suggestion for using the Cochrane Handbook for the conduct of this review and have now included reference to this in our Methods section (Page 10, lines 474-476). The review will be conducted in accordance with Higgins' and Thomas' handbook (version 6.1, 2020) and will be reported in accordance with PRISMA guidelines. Clarified.

5. Participants: Will the authors only include studies where children are reporting on current abuse? Or will you include studies where children report on incidences of abuse in the past? If the latter than I suggest extending the upper age limit to include young adults (up to 25) so that studies that collect retrospective reports of relatively recent childhood experiences might be included.

The inclusion criteria for this review included only prospective studies on child maltreatment, not retrospective studies. There are three reasons for this. First, retrospective studies of child

maltreatment bring in the issue of recall bias which may skew the accuracy of data (Baldwin et al., 2019). Second, the WHO defines child maltreatment as ‘the abuse and neglect that occurs to children under 18 years of age’ therefore we chose to strictly include studies with participants between the ages of 0-18 years (United Nations 2018). Thirdly, one of the aims of this review was to establish the effects the Protection of Children from Sexual Offences Act (POCSO Act) which was established in 2012 by the Indian government to improve reporting of child maltreatment. The POCSO Act is clear in the definition of children as an individual under the age of 18 years. Therefore, to help determine the effect of POCSO on the prevalence of child maltreatment reporting post-2012, we will use studies with children between the ages of 0-18 years to allow accurate comparisons and inferences to be drawn. Therefore, yes, we will include children reporting on current abuse and any abuse occurring during childhood (page8/9, lines 395-410).

6. I would also like to know why retrospective studies of CSA will be excluded – is there a particular reason for this?

Often, prospective and retrospective measures of child maltreatment do not always identify the same individuals (Baldwin et al., 2019). Retrospective studies of child maltreatment are often regarded as a shortcut as significant time; finance and resources are not required to undertake cohort studies (Teicher et al., 2016). There is low agreement between prospective and retrospective assessments of child maltreatment that can be explained by factors such as motivation of reporting, measurement features and memory bias. Firstly, motivation can reduce agreement if individuals may gain something by intentionally withholding data about their maltreatment for embarrassment or fear or feeling uncomfortable or upset. Secondly, child maltreatment assessments are known to have imperfect test-retest reliability (Colman et al., 2016) and prospective measures might only account for the more severe of cases whilst retrospective ‘true positive’ case (Baldwin et al., 2019). Lastly, memory bias can reduce agreement as a result of impaired brain development (Eichenbaum 2017), deficit in memory consolidation of traumatic events or disrupted sleep patterns (Rasch et al., 2013). Therefore, given the variation, the focus on prospective recall (rather than retrospective) of child maltreatment for systematically reviewing the literature on abuse and neglect in children would reduce variation errors, measurement bias and memory bias (Baldwin et al., 2019).

The authorship team will fully ensure that this point on retrospective versus prospective recall of child maltreatment is fully discussed in any subsequent publications resulting from this review and analysis. Amends made (page8/9, lines 395-410)

7. What will the authors do with studies that include both children/adolescents and adults? E.g. 15-25 year olds?

The inclusion criteria for this review will strictly include children aged between 0-18 years as per the definition of child maltreatment by the WHO. If research studies include both children and young adults with ages above 18 years, these studies will be excluded as not having met the inclusion criteria, particularly if it is difficult to extract child-only relevant data from these papers (e.g. lack of subgroup descriptive and analysis for children only).

8. Outcomes: Some definitions of CM include witnessing domestic violence. Can you consider including this?

Our definition of child maltreatment was drawn from two sources: the World Health Organisation definition of child maltreatment which includes all types of abuse and neglect (WHO 2018) and the National Society for the Prevention of Cruelty to Children (NSPCC 2017). The WHO guidelines on early childhood adversities do include witness domestic violence and specifically, Did you see or hear a parent or household member in your home being slapped, kicked, punched or beaten up? when using the adverse childhood experiences international questionnaire. Domestic violence is an important and separate early childhood adversity that exists and occurs at a family level, beyond an individual or child level. We refer to Sidebotham and Heron’s socioecological model of child maltreatment (2001; 2006) to delineate between individual and direct child maltreatment such as

abuse and neglect, and secondary or family level factors such as witnessing domestic violence, witnessing substance misuse or parental incarceration or parental episodes of mental health disorders. As a result, given the inconsistency in the literature in this area, the study authors will adhere to the strict definitions of child maltreatment as referred to in the manuscript by the World Health Organisation and the NSPCC.

Search:

9. Overall, I do not think that the search strategy is adequate and I suggest that the team consult an information retrieval specialist to improve the search strategy. Obvious terms are missing such as “non accidental injury”, “child maltreatment”. An example search that the team might consider using, developed and tested by an info specialists, is available here (lines 1-20)

<https://www.ncbi.nlm.nih.gov/books/NBK385397/>

We thank Dr Hanratty for these comments on ways in which we may improve our search strategy. We have already included the term ‘child maltreat*’ in our search strategy as evidenced by Table 1 and will definitely include the term, ‘non-accidental injury’ to our search strategy (revised table 1). To further clarify, the search strategy presented in the protocol paper was devised in a systematic manner. Firstly, initial search terms were developed with reference to the key words of major publications on child maltreatment and childhood adversities in India. Secondly, we also included the previously peer-reviewed and published MeSH terms and keywords used by the most up to date review in the area of maltreatment in Indian children (but focussed only on child sexual abuse). This published systematic review was the work of our senior author, Dr Vikas Choudhry, who further advised on specific ways in which the search strategy could be adapted and improvised for this specific review. Third, we then created a table of key search terms and keywords which was then circulated among the expert advisory panel as detailed in our protocol paper (Appendix 1) to further improve and enhance the search strategy. And lastly, we did seek expert advice on our search strategy for our preliminary searches and subsequent search Table 1 from a subject librarian advising on systematic reviews based the University of Bristol (Ms. Sarah Herring, Subject Librarian, Bristol Medical School Population Health Sciences).

We will, however, consider ways in which our search strategy may be further improved using additional resources as suggested by Dr Hanratty.

10. The Cochrane library predominantly consists of intervention reviews and RCTs and is unlikely to contain reviews or primary studies relevant to this review. I suggest searching databases specifically for relevant Systematic Reviews (the search strings developed by Sandieson et al are excellent <https://sites.google.com/view/pearl-harvesting-search/ph-synonym-rings/structure-or-study-design/systematic-reviews> In addition to searching the databases already listed in the protocol more fruitful systematic review databases for this topic would be the Campbell Library/Journal and JIB <https://journals.lww.com/jbisrir/pages/default.aspx>

The authorship team will certainly consider addition databases specific to systematic reviews including the Campbell Library/Journal. For clarity, our preliminary searches indicate that the Cochrane library does contain references and papers relevant to our research question and search strategy (n=157) and potentially relevant to our review. The authorship team will consider additional sources such as the Campbell Collaboration, Campbell Systematic Reviews and JBI as part of our extended search strategy. (Page 10, Line 482)

11. PsychIndex – is this an error? Presumably, the authors mean PsycInfo?

Yes, this is correct, and the error has now been rectified throughout the manuscript.

12. The team have not indicated that they intend to search for grey literature which is major omission. I suggest adding, at a minimum, web searches to identify grey literature sources such as government reports, NGO data, international surveys that include India etc. I see later in the discussion that this omission is due to limited resources. I suggest that this makes this work a rapid or restricted review

rather than a systematic review.

This is correct. As a result of a lack of funding and further resources for this project, we were unable to extend the review to include additional resources such as OPEN Grey, government reports and international surveys. However, it should be noted that all government funded studies that included any type of quantitative information that was published in a peer-reviewed journal or as an academic paper, will be included in our systematic review, for example, Kacker's study (2007) was originally released as a government report produced by the Indian Ministry of Women and Child Development. However, given the impact and important of including evidence from the grey literature in systematic reviews (Paez 2017), we will endeavour to include these sources so that they may be introduced into our manuscript discussion and particularly, our narrative synthesis of these important area of research. (Page 16, Lines 686-689 and reference included). We have also amended our title to a rapid review considering this limitation to the current work.

However, we would like it noted in our response to the reviewers that this is an important topic that requires further funding and enquiry, and this work is a systematic review of the published literature undertaken without any specific allocated funds or resources. Therefore, we consider this a significant achievement in advancing the topic area of child maltreatment in India and hope that we may use this to build a narrative in this research area so that we may continue to advance study.

13. Data Management: The data management process is not sufficiently clear in my opinion. Will dual independent screening be done at title/abstract screening stage and full text? Recording reasons for exclusion and inclusion for all studies is not typically necessary; instead I suggest reasons for exclusion only be recorded for articles screened at full text stage.

We can see Dr Hanratty's perspective on our section on data management. The authorship team were able to elaborate on our data management process in lines with the suggestions above. To clarify, dual independent screening will be done at the title/abstract screening stage with the use of Rayyan (blinded assessments) and coding using their traffic light system for including, excluding or discussing relevant studies based on title, abstract information and year of publication. It will also be done by the first authors and secondary authors on any selected full text articles. However, in order to ensure that studies were included or excluded at the title/abstract screening stage, they will be coded or categorised with tags such as 'wrong outcome' or 'wrong population', in order for the authors and review team to openly discuss the reasons for any discrepancies at the preliminary stage. This will be recorded in our Rayyan files.

We will also ensure that the reasons for exclusion will be recorded for articles screened at the full text stage and this be presented in the final manuscript. We have clarified this in our manuscript with a reference [Page 11, lines 526-534].

14. Quality assessment: This section conflates screening quality with assessment of quality and risk of bias in included studies. Please separate the two. The quality assessment section should only refer to how risk of bias/quality of the studies included in the review will be assessed. The Newcastle Ottawa Scale (NOS) was primarily designed to assess quality of non-randomised studies of outcomes and NOT cohort/cross-sectional studies of prevalence. Adapted versions of the scale are available and should be considered. Please document any adaptation to the scale used.

We will assess the risk of bias and the quality of studies included in this review using the Newcastle-Ottawa Quality Assessment scale (NOS) as this considers quality in three domains: selection, comparability and outcome assessment. This has been recommended for use by the Cochrane Collaboration to assess studies for inclusion into a systematic review. We should have been clear as to which NOS format we would be using. As per our example (Appendix 3) which was retrieved during a preliminary literature search, we will be using the NOS tool for cross-sectional studies which was adapted from the NOS for cohort studies as created and credited to Herzog and colleagues (2013) in our protocol paper. We have now made changes to this section to separate screening quality with risk of bias and quality assessment and included details and reference to the adapted version of NOS.

15. Can the authors state why they will exclude studies written in languages other than English? Will the impact of this decision be discussed in the review?

The authorship team considered including studies in all languages and Indian dialects but decided to include only English papers. Firstly, the Cochrane Handbook acknowledged the risk of bias in reviews containing exclusively English language studies and recommends assessing this on a case-by-case basis (Higgins et al., 2011). The inclusion of non-English papers is also met by the pragmatic concerns of search specialised non-English language databases and this is further limited by the authorship's language skills; resources available to pay for access to specialist databases and professional translators. We will ensure that any repercussions of this decision will be discussed in the final published review.

Our expert panel includes research and academic experts in the field of paediatrics and child maltreatment in India and on discussion with the team, we did a preliminary search on peer-reviewed published articles in Hindi and other regional languages. Any potential sources of bias in our findings because of language limitations will be discussed in the final manuscript.

Also might be interesting to see if some of the questionnaires used in the study had been validated for India in local language and discuss that too in potential sources of bias!

16. Discussion: Typically, a protocol would not include a discussion and some of the information here belongs in the methods section, specifically

Patient and public involvement: I do not agree that it is not appropriate for the public to be involved in the conceptualization of systematic reviews.

We appreciate Dr Hanratty's comment on the discussion section of our protocol paper. We followed the guidance provided by BMJ Open on what should be included in a discussion section of a protocol paper. We have not included any new information and rather make the case for why this particular review is timely and important and lastly, what the implications may be for policy and research in India with a considerable section on the strengths and caveats of our proposal. The inclusion of researchers, clinicians, academics and policymakers was considered when creating our expert advisory panel on this topic. Furthermore, the inclusion of our 'patients of interests' i.e. children with experiences of adversities and specifically, child maltreatment, would be unethical and at this stage of evidence synthesis in a systematic review be redundant. We can assure Dr Hanratty that the inclusion of child psychologists and experts in the field of childhood adversity would represent sufficient breadth and depth of expertise so as to get this pressing review conducted and off the ground.

17. PRISMA: While the authors have filled out the PRISMA checklist the protocol does not actually contain enough detail to meet PRISMA standards. E.g. Was a protocol already produced with PROSPERO? The authors have not specified eligibility criteria in enough detail.

The PRISMA checklist was rightly highlighted as a publication guide for any future systematic review manuscripts (earlier comment). However, we acknowledge the points raised and have specified the eligibility criteria in more detail in the methods section of our manuscript. These changes can be found here, page 8, lines 382-417.

Specifically, are there limits on the year the study was conducted?

This review will include all research studies between 1st January 2005 – 1st January 2020. The reason for this time window is to establish a clear 8-year period pre- and post- 2012, which was the year the Protection of Children from Sexual Offences Act (POCSO Act) was introduced to allow accurate comparison and inferences to be drawn. We have now ensured that this timeline is explicitly mentioned in the Methods (Page 8, line 383-384).

18. What will you do with studies including participants with an overlapping age range? Information sources are not described with dates, errors are present and potentially useful databases are not included (e.g. CINAHL, Web of Knowledge SSCI, Campbell Library, JBI). The search is not

adequately developed or reported.

If research studies include participants with an overlapping age range, as long as these age ranges are below the age of 18 years, the information will be included into the review. We would attempt to extract specific information from the individual studies or from supplementary information published by the authors. However, to confirm, these studies will be included in our rapid review as long as they meet all other inclusion criteria set out.

19. I commend the team on a good idea for a review that is needed and hope that my comments are helpful in improving the quality of the work.

Thank you Dr Jennifer Hanratty, for your time in providing valuable feedback. This review aims to provide data on the magnitude, trends, gendered and urbanization characteristics of a country home to 20% of the world's children therefore we agree it is definitely needed and necessary to help with future child protection policies and legislation.

Reviewer: 3

Dr. David Walsh, Glasgow Centre for Population Health

Comments to the Author:

Thanks for the opportunity to comment on the protocol for a systematic review of child maltreatment in India.

Overall I found the protocol to be helpfully detailed in terms of its rationale and proposed methods. My comments, therefore, are all relatively minor, with the sole aim of hopefully adding some additional clarity to a small number of aspects.

Response to Reviewer 3, Dr David Walsh:

Thank you, Dr David Walsh, for the constructive feedback. The study protocol was designed to be thorough and detailed with the aim of providing high quality research and avoiding bias or errors. We agree your comments have enabled us to make relevant changes and improvements to the manuscript. We have also now provided further justifications, as per your comments, to improve the overall quality of our protocol paper.

The comments are as follows:

1. Other than a passing mention of confounders in the data collection Table (Table 2), there is no real discussion of poverty/socioeconomic position, which I found rather surprising. Given the clear understanding of the role of poverty in increasing risk of maltreatment (e.g. see the JRF review from 2015 among other papers), I would have thought this would at the very least have been flagged up as a key issue in the review i.e. the need to collect such data to aid interpretation of results (including of any meta-analysis).

We acknowledge and understand well this comment from Dr Wash and admit there could and should have been more detail on the socioeconomic factors, particularly given the fact that we are undertaking this review in a WHO defined developing economy and lower-and middle-income country. We have therefore sought to properly rectify this oversight by adding details to our introduction (page 4, lines 150-171) and our discussion section (page 15, lines 631-634).

2. It would also be helpful to justify why the authors are only looking at papers published between 2005-2020. I suspect (?) it relates to the date of the child sexual abuse legislation that is mentioned, but it's not clear. As a 'by the by', the inclusion criterion is stated in the abstract, but not (I don't think?) in the actual methods, and obviously should be (although apologies if I have just missed it!)

This review will include all research studies between 1st January 2005 – 1st January 2020. The reason for this time window is to establish a clear 8-year period pre- and post- 2012, which was the year the Protection of Children from Sexual Offences Act (POCSO Act) was introduced to allow

accurate comparison. As this comment was raised by Reviewer 2 as well, we have now made this information clear and available in the manuscript (page 8, line 383-384).

3. Additionally regarding dates, reviewers of protocols are specifically asked to check that the dates of when the study is being/will be undertaken are included, so these could also be helpfully added. We agree with Dr Walsh on this suggestion. As mentioned in the reply to Comment 2 above, the specific dates of the study have now been added and justified in line (page 8, line 383-384)).

4. The methods also state that retrospective studies will be excluded. Given that quite a few studies in the literature are studies of adults asked about maltreatment in childhood, it would be good just to justify/explain this exclusion.

We thank Dr Walsh for raising this point that has already been addressed previously (similar comment by reviewer 2, Dr Hanratty, page 8, lines 395-410). Our response was:

Often, prospective and retrospective measures of child maltreatment do not always identify the same individuals (Baldwin et al., 2019). Retrospective studies of child maltreatment are often regarded as a shortcut as significant time, finance and resources are not required to undertake cohort studies (Teicher et al., 2016). There is low agreement between prospective and retrospective assessments of child maltreatment that can be explained by factors such as motivation of reporting, measurement features and memory bias. Firstly, motivation can reduce agreement if individuals may gain something by intentionally withholding data about their maltreatment for embarrassment or fear or feeling uncomfortable or upset. Secondly, child maltreatment assessments are known to have imperfect test-retest reliability (Colman et al., 2016) and prospective measures might only account for the more severe of cases whilst retrospective 'true positive' case (Baldwin et al., 2019). Lastly, memory bias can reduce agreement as a result of impaired brain development (Eichenbaum 2017), deficit in memory consolidation of traumatic events or disrupted sleep patterns (Rasch et al., 2013). Therefore, given the variation, the focus on prospective recall (rather than retrospective) of child maltreatment for systematically reviewing the literature on abuse and neglect in children would reduce variation errors, measurement bias and memory bias (Baldwin et al., 2019).

The authorship team will fully ensure that this point on retrospective versus prospective recall of child maltreatment is fully discussed in any subsequent publications resulting from this systematic review and analysis.

Amends made (pg8, lines 395-410).

5. Non-English language papers are also excluded. Although understandable, it would be helpful – particularly for a study of maltreatment in India – to say a little bit more about that (again in terms of justification). As the authors will know, the issue of 'Anglophone bias' is often highlighted, and your study is one where I would have thought careful and clear justification for excluding non-English papers would be particularly helpful.

This review included English Language research papers only. We appreciate the importance of acknowledging the limitation this creates, particularly because the subject matter of child maltreatment prevalence rates is being studied in India - a country home to several hundred languages. Excluding non-English papers also risks missing key data and potentially introducing bias. Furthermore, we unfortunately did not have access to adequate funding enabling access to resources (e.g. professional translator) to help with analysing non-English papers. We take your points on board and have therefore added this justification to our limitation section (page 16, lines 686-689) and hope to gain access to professional translators when working on future papers, if we are able to secure funding to continue and develop this research area.

6. In terms of the meta-analysis, the authors may find it helpful to think about (and discuss) how any such analysis might deal with profound differences in measurement across different studies e.g. of age of child (clearly important in terms of potential exposure to maltreatment), or of the number and types of maltreatment recorded (e.g. comparing studies of sexual abuse alone, with studies of

multiple forms of maltreatment).

We thank Dr Walsh for this insightful comment which is certainly worth discussing as a team and deliberating as part of any subsequent meta-analysis. As part of our data extracting process, we will be looking at specific types of child maltreatment including the prevalence rates recorded over a one-year period and 'over a lifetime' of a child. We will be recording the age of participants included in the research and also discuss the strengths and caveats of different measurement tools used to assess child maltreatment. All of this information will be extracted, and we may use this to create sub-groups of analysis or risk-groups of the paediatric population to study with more granularity. Amends made on page 14, lines 595-601.

7. There is no mention of whether or not any grey literature will be included in the review, and it would be useful to be clear about this also.

This is correct. As a result of a lack of funding and further resources for this project, we were unable to extend the review to include additional resources such as OPEN Grey, government reports and international surveys. However, it should be noted that all government funded studies that included any type of quantitative information that was published in a peer-reviewed journal or as an academic paper, will be included in our review, for example, Kacker's study (2007) was originally released as a government report produced by the Indian Ministry of Women and Child Development. However, given the impact and important of including evidence from the grey literature in systematic reviews (Paez 2017), we will endeavour to include these sources so that they may be introduced into our manuscript discussion and particularly, our narrative synthesis of these important area of research (page 16, Lines 686-689 and reference included).

8. Finally, this is a review of child maltreatment, as is clear from both the title and stated study design outcomes. However, a large chunk of the introduction discusses the much broader concept of adverse childhood experiences (ACEs) – and I'm not sure how helpful that is. As all the authors will know well, maltreatment accounts for around half the often-used collective measure of 10 ACEs, with the other half of the latter being a much broader set of measures of (unhelpfully titled) "household dysfunction". Although ACEs have obviously been the focus of a huge amount of research in recent years (see critique by Kelly-Irving and Delpierre regarding reasons why), they have also subject to considerable criticism for a number of reasons. There is also a large, separate, literature on maltreatment which – again as you will know – predates much of the ACEs research. I think it would be clearer, therefore, just to focus on maltreatment alone, without the potentially confusing mentions of ACEs (other than stating that maltreatment is included within some larger questionnaires including those examining ACEs). For the same reason, I'm not sure the term "maltreatment ACEs" is particularly helpful in that regard.

We acknowledge this comment and discussion by Dr Walsh and concur that this is an area of research and enquiry which predates the popularisation and discourse on 'Adverse childhood experiences' (ACEs) by Felitti and Anda. The first mention of this phrase in our protocol manuscript was under 'An Indian Context' in the introduction of the paper and I've now revised this to refer specifically to the title, scope and context our research proposal i.e. focus on child maltreatment. We have therefore re-written this section to be more inclusive and cognizant of other works in the research area, alternative ways of thinking of childhood adversity and applying this directly to what we know from an Indian context. We trust the changes we have made are considerate of Dr Walsh's suggestion and have removed all mention of the term maltreatment ACEs, thereby removing any ambiguity or conflating two constructs. These changes can be found in the tracked version, page 4 (line 173) and page 5 (line 189-214) and included updated references.

Finally (no. 2), but most importantly, a systematic review is often a hard shift – so I wish you luck with what is clearly a very worthwhile and important endeavour.

We thank Dr Walsh for his comments, advice and indeed hope that the project proves fruitful and

productive in the coming years.

Reviewer: 1

Competing interests of Reviewer: none

Reviewer: 2

Competing interests of Reviewer: None declared

Reviewer: 3

Competing interests of Reviewer: None declared

VERSION 2 – REVIEW

REVIEWER	Walsh, David Glasgow Centre for Population Health
REVIEW RETURNED	14-Jun-2021
GENERAL COMMENTS	Thanks for responding in such detail my original review - and, therefore, in making the suggested changes. Good luck with the review!

VERSION 2 – AUTHOR RESPONSE

Thank you for this opportunity to work on the paper and to Reviewer 3 for their well wishes.